# Detection of Breast Cancer Cells Using Acoustics Aptasensor Specific to HER2 Receptors

**DOI:** 10.3390/bios9020072

**Published:** 2019-05-27

**Authors:** Alexandra Poturnayová, Ľudmila Dzubinová, Monika Buríková, Jozef Bízik, Tibor Hianik

**Affiliations:** 1Institute of Animal Biochemistry and Genetics, Center of Biosciences SAS, Dúbravská cesta 9, 840 05 Bratislava, Slovakia; alexandra.poturnayova@savba.sk; 2Department of Nuclear Physics and Biophysics, Faculty of Mathematics, Physics and Informatics, Comenius University, Mlynská dolina F1, 842 48 Bratislava, Slovakia; ludmila.valigurska@gmail.com; 3Cancer Research Institute, Biomedical Research Center SAS, Dúbravská cesta 9, 840 05 Bratislava, Slovakia; monika.burikova@savba.sk (M.B.); jozef.bizik@savba.sk (J.B.)

**Keywords:** breast cancer cells, HER2 receptors, DNA aptamer, acoustic biosensor, gold nanoparticles

## Abstract

Detection of the breast cancer cells is important for early diagnosis of the cancer. We applied thickness shear mode acoustics method (TSM) for detection of SK-BR-3 breast cancer cells using DNA aptamers specific to HER2 positive membrane receptors. The biotinylated aptamers were immobilized at the neutravidin layer chemisorbed at gold surface of TSM transducer. Addition of the cells resulted in decrease of resonant frequency, f_s_, and in increase of motional resistance, R_m_. Using gold nanoparticles (AuNPs), modified by aptamers it was possible improving the limit of detection (LOD) that reached 550 cells/mL, while without amplification the sensitivity of the detection of SK-BR-3 cells was 1574 cells/mL. HER2 negative cell line MDA-MB-231 did not resulted in significant changes of f_s_. The viability studies demonstrated that cells are stable at experimental conditions used during at least 8 h. AuNPs were not toxic on the cells up to concentration of 1 μg/mL.

## 1. Introduction

HER2 positive breast cancer is among most aggressive type of disease with low percentage of survival. However, substantial progress in early stage diagnostics together with progressive methods of therapy substantially improved the situation. Nevertheless, despite of the current achievements, the diagnostics of the breast cancer is rather difficult considering that at the first stage it is usually without significant symptoms. Most of the patients are in general diagnosed in late stage of this disease. The main reason is lack of the effective early stage diagnostics tests. For the purpose of diagnostics of HER2 positive breast cancer various methods have been developed. However, approx. 20% of the tests can be casual.

The epidermal growth factor receptor (EGFR) is at the front of the signaling cascade, that affect the cell proliferation and differentiation [1]. This include HER2 receptors that are overexpressed by 15–30% in breast cancer. This oncomarker is therefore used as an indicator for decision in the therapy as well as the therapeutical target for breast cancer. The breast cancer patients have increased concentration of HER2 in a blood at the range of 15–75 ng/mL in comparison with healthy individuals (2–15 ng/mL) [2]. For HER2 positive breast cancer is typical rapid growth of tumor, low degree of survival and better response to adjuvant therapy [3]. Increased expression of HER2 has been also observed at other types of cancer such as stomach, ovarian, urocyst, lung and endometrial cancer [4]. Despite of the progress in diagnostics based on detection of HER2 receptors, approx. 20% of the tests are inaccurate. Therefore the American Society of Clinical Oncology (ASCO) as well as the American Society of Clinical Pathology (ASCP) recommended two tests for HER2 oncomarker: immunohistochemical method (IHC) and fluorescence in situ hybridization (FISH). However, these tests are time consuming and require qualified staffs. In addition the risk of false-positive results is in the range of 20–50% [5]. The biosensor technology can overcome existing problems. The biosensor consist of three main parts—the sensing element (provided recognition of the analyte), transducer (transduces chemical signal into the measurable physical signal—electrical, optical, etc.) and analytical instrument for evaluation of the sensor response. Important part of the sensor is the sensing part that is composed of corresponding receptor immobilized at suitable surface [6]. Among receptors the DNA/RNA aptamers are of increasing interests. Aptamers are prepared in vitro by combinatorial chemistry based on the SELEX method (Systematic Evolution of Ligands with Exponential enrichment) [7,8]. The aptamers play increasing role in the therapy and diagnostics of the cancer diseases. They are single stranded DNA or RNA oligonucleotides known also as “chemical antibodies” [9]. Aptamers in solution fold into defined secondary and tertiary structure forming binding site for specific analyte (low molecular compounds, macromolecules, cells and tissues). It is known that they bind the analyte with high selectivity that is comparable with those for antibodies. Moreover, in contrast with antibodies the aptamers are more stable in respect of temperature denaturation-renaturation and the sensor based on aptamers can be regenerated in certain conditions, such as for example high ionic strength [10]. The receptors that are overexpressed in cancer cells (for example HER2 or protein tyrosine kinase 7 (PTK7)) are the main targets of the aptamers in the therapy. So far large number of aptamers for targeting cancer cells have been developed. HER2 receptor is considered as an important biomarker due to its key role in the progression of aggressive type of the breast cancer. During recent years many studies are focused on the development of aptamers for HER2 receptors for the therapeutics and diagnostics purposes [4]. For example Gijs et al. [11] analyzed two aptamers HeA2_1 and HeA2_3 that targeted the adherent HER2 positive cancer cell. Both aptamers bound with a high specificity to the cell lines SKOV3 and SK-BR-3, that overexpressed HER2 receptors. The application of fluorescently labeled aptamers revealed increased fluorescence following the interaction with above mentioned HER2 positive cancer cells, while no changes in the fluorescence took place for control cells MDA-MB-231 that do not contain HER2 receptors. Li et al. [12] selected DNA aptamer LXL-1-A, that bound with a high specificity to the metastasis and breast cancer tissue in 76% cases of patients. Liu et al. [13] selected HB5 DNA aptamer that targeted HER2 protein. This aptamer recognizes extracellular domain (ECD) of HER2 protein. They found that this aptamer bind with high specificity to HER2 in comparison with other blood proteins such as albumin and trypsin. The binding studies of various HER2 positive and negative cell lines revealed that HB5 aptamers bind strongly to the HER2 positive cell line SK-BR-3 but only weakly to the HER2 negative MDA-MB-231 and MCF-7 cell lines.

Application of the gold nanoparticles is rather useful for cell and tissue imaging as well as for visualization of the targeted drug delivery by means of aptamers [14]. For the diagnostics purposes Medley et al. [15] prepared gold nanoparticles (AuNPs, diameter 20 nm) modified by DNA aptamers with high affinity to the CCRF-CEM cell lines. These aptamers conjugated with AuNPs revealed high sensitivity and selectivity to the cancer cells. These complexes were adsorbed at the membranes of the cancer cells that allowed their better visualization due to unique spectral properties of the gold nanoparticles.

Among biosensors those based on acoustics principles are of special interest. They allowing study in real time the processes connected with cell adhesion, cell morphological changes, apoptosis as well as detection of the cells based on receptors specific to the cancer markers in their membranes [16]. They also allowing fast and relative easy detection of the affinity interactions at surfaces [17]. Elmlund et al. [18] developed acoustics cell sensor that was able to detect interaction between trastuzumab antibodies and the SKOV3 ovarian cancer cell lines. SKOV3 cells have been incubated at specially designed chips COP-1 (cell optimized polystyrene) during 24 h. They showed that the incubation as well as density of adsorption of the cells affect the interaction with antibodies. It has been also shown that thickness shear mode acoustics sensor (TSM) is convenient tool for obtaining information on the recognition of the leukemics cancer cells [19]. The transformation of normal cells into cancer cells is accompanied by the changes of the cell morphology and the rearrangement of the cytoskeleton. These changes affect the viscoelastic properties of the cells as well as cell adhesion properties. Zhou et al. [16] compared dynamics adhesion of the normal human epithelial (HMEC) and breast cancer cells (MCF-7) in real time. From the ratio of the changes of the resonant frequency and dynamic resistance they determined cell dynamic viscoelastic index (CVI). CVI for the MCF-7 cell line has been 2.5 times lower in comparison with those for HMEC. In our recent work [19] we showed that TSM method is rather useful for detection leukemia Jurkat and MOLT-4 cell lines containing protein tyrosine kinase 7 (PTK7) receptors based on DNA aptamers specific to PTK7. In this work we applied for the first time this method for detection HER2 positive breast cancer cells.

## 2. Materials and Methods

Human breast cancer cells SK-BR-3 (HER2 positive) and MDA-MB-231 (HER2 negative, PTK7 positive) were incubated in Dulbecco‘s modified Eagle’s medium (DMEM) (PAN Biotech, Aidenbach, Germany), supplemented with 10% fetal bovine serum (FBS, PAN Biotech) and 1% penicillin-streptomycin (Sigma Aldrich, Schnelldorf Germany). Cell cultures were maintained at 37 °C (human body temperature) in humidity within a 5% CO_2_ atmosphere. All experiments were performed on cells in the exponential growth phase. Cell lines were obtained from the Cancer Research Institute (Biomedical Research Center of the Slovak Academy of Sciences, Slovakia). All cell lines were screened for contamination by mycoplasma.

Neutravidin (NA) was purchased from Thermo Fischer Scientific GmbH (Dreieich, Germany). Inorganic salts, ethanol, ammonia, hydrogen peroxide (p.a.) were from Slavus (Bratislava, Slovakia). Trypan blue for microscopy and trypsin were purchased from Sigma Aldrich. The DNA aptamers modified at its 5’-end by biotin were purchased from Eurogentec (Seraing, Belgium). The aptamer sequences (Table 1) were taken from [11,13,20,21]. The aptamer stock solutions were prepared by dissolving lyophilized oligonucleotides in TE buffer (1 mM EDTA, 10 mM Tris, pH 8). 20 nm gold nanoparticles (AuNPs) conjugated by streptavidin (StpA-AuNPs) were purchased from Nanocs (New York, USA). As a buffer we used PBS (10 mM phosphate buffer saline: 10 mM NaH_2_PO_4_, 1.8 mM KH_2_PO_4_, 137 mM NaCl and 2.7 mM KCl, pH 7.4). The deionised water was prepared by PureLab Classic UV (Elga Water Systems, High Wycombe, UK).

### 2.1. Preparation of the Cells and Their Viability

The breast cancer cells SK-BR-3 (HER2 positive) and control MDA-MB-231 cells (HER2 negative) were three times washed with 2 mL of PBS, and then incubated with 1 mL of trypsin at 37 °C for 5–10 min. FBS was then added to quench the proteinases. After washing with PBS buffer, the trypsin treated cells were used for experiments. Before the experiments it was necessary to prepare samples with known concentration of the cells.

For this purpose we used the method based on direct cell counting in a Bürker cell (Paul Marienfeld GmbH & Co. KG, Lauda-Königshofen, Germany). In further experiments the undamaged cells were counted and diluted for desired concentration in the range of 50–5 × 10^5^ cells/mL.

The cells were added into the 24 wells plate with a concentration of 5 × 10^4^ cells/well. Subsequently the cells were incubated during 72 h in an incubator (ESCO CelCulture^®^ CO_2_, Singapore). Then the growth medium was removed and the cells were washed twice in a PBS. Cells in PBS were added into the wells. The concentration of the cells has been determined after 1, 2, 4, 6, 8 and 10 h in PBS following their incubation at various temperatures: 37 °C in the incubator, at ambient temperature 20 °C and at 4 °C in refrigerator [22,23].

The cell viability was based on a trypan blue test. The trypan blue does not penetrate through the plasmatic membrane of undamaged cells, while the damaged cells are blue colored [22,23]. The trypan blue (0.4% stock solution) has been diluted with cells in the culture medium DMEM in a weight ratio 1:1, so the concentration of the dye was 0.2%.

### 2.2. Cell Proliferation Study at Presence of Gold Nanoparticles

In one series of experiments we used gold nanoparticles (AuNPs) for amplification of the cell detection. For this purpose it has been necessary to analyze the cytotoxicity of the nanoparticles. We used AuNPs conjugated with streptavidin (StpA-AuNPs) in a concentration range 1–10 μg/mL. Cytotoxicity has been analyzed in the special incubator IncuCyte ZOOM™ kinetic imaging system (Essen BioScience, Ann Arbor, MI USA). The breast cancer cells in a concentration 10^4^ cells/mL have been added into the 24 well plate and incubated during 72 h. Then the cultivation medium has been removed, and after cell washing replaced by fresh DMEM medium without FBS, or alternatively instead of DMEM the PBS has been used. Subsequently the AuNPs have been added in certain concentration.

### 2.3. Thickness Shear Mode Method

The thickness shear mode method (TSM) is thanks to its sensitivity one of the most used acoustics methods in biosensing and for surface characterization. It is based on the measurement of the series resonant frequency, f_s_, and motional resistance, R_m_. While the f_s_ value reflects the changes in mass or thickness of the sensing layer at the surface of AT-cut quartz crystal, the R_m_ value is responsible for the viscosity contribution [24]. The experimental setup consisted of a vector network analyzer 8712ES (Agilent Technologies, Santa Clara, CA, USA) connected to 8 MHz AT-cut quartz crystal (Total Frequency Control Ltd., Storrington, UK). The crystal was covered at both sides by gold electrodes. The area of the central part of the electrode that served for preparation of the sensing layer was 0.2 cm^2^. The crystal has been mounted into the acryl flow cell. The cell was the generous gift of Professor M. Thompson (University of Toronto, Canada). The construction of the cell can be found in the paper by Tassew and Thompson [24]. The digital syringe pump Genius Plus (Kent Scientific, Torrington, CT, USA) has been used for the addition of buffers and cells onto the crystal surface with a constant flow rate of 50 μL/min.

### 2.4. Preparation of the Aptasensor

Prior to the preparation of the sensing layer at one side of the quartz crystal, it was necessary to accurately clean the gold layer. For this purpose we used chemical cleaning by basic Piranha solution. The crystal has been immersed into the hot mixture (70 °C) of basic Piranha, composed of ammonium (NH_3_), water and hydrogen peroxide (H_2_O_2_) in a volume ratio 1:5:1 for 25 min followed by rinsing in MilliQ water. This process was repeated three times. The crystal was subsequently washed by copious volumes of distilled water, then by 90% ethanol, and finally dried in the stream of nitrogen gas. The clean crystal has been then mounted into the flow cell and was ready for further modification. For this purpose the water solution of neutravidin (NA) in a concentration of 125 µg/mL has been allowed to flow to the surface of the gold layer of the quartz crystal.

Thanks to the chemisorption (presence of SH groups at the neutravidin molecules) a self-assembled NA layer was formed. The steady-state conditions, where both the resonant frequency and motional resistance become stable, has been reached after 35 min following starting of the NA injection. Because the DNA aptamers were diluted in the PBS, the NA surface was first cleaned by distilled water and then by PBS. Then the biotin-modified DNA aptamers in a concentration of 0.25 μM have been allowed to flow during 10–15 min until the resonant frequency and motional resistance reached steady-state values. Due to high affinity of biotin to NA a stable aptamer layer was formed. The sensing surface was then washed by PBS in order to remove physically adsorbed aptamers. Because the cells were incubated in a DMEM cultivation media, the surface of the sensing layer was washed by this solution. The sensing layer has been then ready for study of the interaction with the breast cancer cells. The scheme of the crystal modification is presented on the Figure 1.

### 2.5. Modification of Gold Nanoparticles by DNA Aptamers

For amplification of the detection of the cancer cells we used gold nanoparticles modified by streptavidin (StpA-AuNPs). The NPs of 20 nm diameter in a concentration of 3.5 × 10^12^ NPs/mL have been used. The NPs were modified by biotinylated HB5 aptamers as follows: 2 µL of NPs in a concentration of 0.5 mg/mL have been mixed with 0.25 μM aptamers in a PBS. The solution has been incubated during 1 h. Then the solution of aptamer-modified NPs has been centrifuged at 13,400 rpm during 35 min. After centrifugation the supernatant with unbounded aptamers has been removed and the pellet was diluted in PBS.

## 3. Results and Discussion

### 3.1. The Study of Cell Viability

Viability is one of the basic parameters in cell culture handling. During measurements, the cells are exposed to adverse conditions (temperature change, lack of culture medium). Therefore, the survival of the cells under these conditions had to be verified. We tested the effect of PBS buffer on the cell viability. To compare the effects of temperature and the concentration of carbon dioxide (CO_2_), cell viability was tested under different conditions—in incubator at 37 °C and 5% CO_2_, at room temperature at 20 °C, and in refrigerator at 4 °C [25]. Figure 2 summarizes the viability of SK-BR-3 cell line in PBS. We observed a higher than 5 × 10^4^ cells/mL in some wells in multiwell plates, which is related to the cell proliferation during settling. After incubation in PBS, the cells could no longer be divisioned.

The lowest cell number decrease was observed at room temperature, but even at 37 °C and 4 °C, there was no significant reduction in the cell number over time compared to the original number (5 × 10^4^ cells/mL). We observed a decrease in the cell concentration by 16% after 8 h incubation, when testing the viability of the SK-BR-3 cells in PBS at 4 °C. After PBS administration we observed a decrease in the cell number by 18% in incubator at 37 °C. From our cell viability results in PBS, we can conclude that the SK-BR-3 cell line is stable under laboratory conditions. Our results are consistent with Han et al. [26]. They also did not observe a decrease in viability of the adherent breast cancer cell line MCF-7 under laboratory conditions.

### 3.2. The Study of the Toxicity of Gold Nanoparticles on the Breast Cancer Cells

The cells in a flow system of TSM do not grow. However, it is crucial to provide viability of the cells during the whole experiment as well as following the addition of gold nanoparticles (AuNPs). It was therefore important to analyze what concentration of AuNPs is not toxic and does not induce cell apoptosis. Verification of the cytotoxic effect of AuNPs is a major factor that predetermines them in biomedical applications. AuNPs have long been considered inert, non-toxic and biocompatible. Several studies were focused on the cytotoxicity of AuNPs under different conditions (temperature, pH, various concentrations and size of AuNPs) and most of them confirmed the non-toxic properties of AuNPs. On the other hand, some studies demonstrated several adverse effects of AuNPs on the cell membranes, mitochondria or nuclei and caused DNA damage, oxidative stress, apoptosis, mutagenesis, etc. [27,28]. Therefore, we performed a proliferation assay on SK-BR-3 cells after adding streptavidin-modified AuNPs in a concentration range of 1–10 µg/mL.

The optimal concentration of StpA-AuNPs (gold nanoparticles conjugated to streptavidin) with a diameter of 20 nm for in vitro experiments was determined by monitoring the cells in the Incucyte incubator. Incucyte allows us to monitor the influence of various factors on the development of adherent tumor cells. We observed the cell confluence in DMEM culture medium or PBS buffer after addition of nanoparticles in the concentration range 1–10 µg/mL to SK-BR-3 (HER2+), MDA-MB-231 (HER2-) and healthy breast cancer cell line MCF10A. Cells were scanned with an Incucyte scanner during 90 h. The result of these measurements for the SK-BR-3 cell line in DMEM are shown in Figure 3.

We observed minimal cytotoxic effect on the cells in both cases (DMEM and PBS) only at a AuNPs concentration of 1 µg/mL. In the case of cells in DMEM medium after 10 h of addition of nanoparticles, the cells began to proliferate again. An opposite effect was observed in PBS. After addition of PBS to the cells, the confluency was reduced by 2.5%, and then after 33 h caused a 1% decrease yet. Thus, we can conclude that for further experiments 1 µg/mL of AuNPs can be considered as an optimal concentration.

Smolkova at al. [29] verified the toxicity of 20 nm AuNPs in the culture medium on SK-BR-3 breast cancer cells. Cells were exposed to four different concentrations of AuNPs (0.8 µg/mL, 4 µg/mL, 21 µg/mL and 28 µg/mL) for 2, 24 and 48 h. The results of this study confirmed the cytotoxic effect of AuNPs already at 4 μg/mL. At a concentration of 0.8 µg/mL no toxic effect was observed. Therefore, we used a concentration of 1 μg/mL of StpA-AuNPs to carry out the experiments, which was tested after modification with the 0.25 μM HB5 aptamer. The StpA-AuNPs experiment modified with the HB5 aptamer (HB5-StpA-AuNPs) was performed in the same manner as for unmodified nanoparticles. The results of the analyses showed that the HB5-StpA-AuNPs complex is not toxic to the cells at the time of observation. Aptamer-modified AuNPs (DNA-AuNPs) can be incorporated into the cells depending on their concentration, incubation time, or cell line. So far, however, no toxic effect of these DNA-AuNPs on target cells has been found [30], which also confirms our findings.

### 3.3. Detection of Breast Cancer Cells by TSM Method

In this part we present the study of the interaction of breast cancer cells with the surface of acoustics aptasensors. Three different aptamers were used for sensor preparation (Table 1). Two of them (HB5 and HeA2_3) were selective to HER2 receptors localized in a membrane of breast cancer cells, while sgc8c aptamer was used as a control. It was selective to PTK7 receptors that are overexpressed in leukemics cells. First, we analyzed all steps of the sensor preparation. The sensor has been prepared by the already approved sandwich method [19] that utulizes the advantage of strong interactions between neutravidin (NA) and biotinylated aptamers. First, the clean gold surface of an acoustic transducer has been modified by NA dissolved in deionised water (concentration 125 μg/mL) in a flow injection mode. As it can be seen from Figure 4 chemisorption of NA resulted in strong decrease of resonant frequency, f_s_, on average by 157.8 ± 27.3 Hz. However, the changes of motional resistance, R_m_, were minimal. This is evidence of the formation of a rigid self-assembled layer of NA at the gold surface [24]. The steady-state conditions where both f_s_ and R_m_ values become stable have been reached after 35 min following starting of the NA injection. Washing of the surface by deionised water resulted in slight increases of f_s_ and R_m_ values. This corresponds to removal of physically adsorbed NA molecules. The NA forms very stable layers at gold surface. This has been demonstrated also by the study of the topography of these layers using atomic force microscopy (AFM). The results are presented in the Appendix A. The next step consisted of immobilization of the DNA aptamers. However, because aptamers were diluted in a PBS and the changes of ionic properties affect the acoustics parameters, the surface of NA layer has been washed by PBS. As it can be seen from Figure 4, this step resulted in sharp changes of both f_s_ and R_m_ values, which agree well with the already discussed effect of the changed ionic conditions on the molecular slip between the sensing layer and surrounding water solution [31]. Addition of biotinylated DNA aptamers (HB5) in a concentration of 0.25 μM resulted in decrease of f_s_ value in average by 57.10 ± 16.03 Hz and in an increase of R_m_ by 4.80 ± 0.76 Ω.

The decrease of resonant frequency is due to the strong binding of biotinylated HB5 aptamers to NA molecules, while the increase of R_m_ is caused by the increase of molecular slip between the sensing layer and the surrounding PBS. This phenomenon has been observed previously in our work [32]. The washing of the surface by PBS resulted only in slight changes of measured values, which is evidence of the removal of physically adsorbed aptamers. After this step it was necessary to wash the sensing layer by PBS with DMEM medium in which the cells were incubated. This step resulted in a further decrease of f_s_ and increase of R_m_. The effect is similar to those caused by changed ionic strength. After this procedure the aptasensor was ready for analysis of the interaction with the cells. Figure 4 shows also typical changes of f_s_ and R_m_ values folowing addition of the breast cancer cells SK-BR-3 at concentration of 5 × 10^4^ cells/mL. It can be seen that addition of the cells in a flow injection mode resulted in the decrease of the resonant frequency and in an increase of the motional resistance. This is evidence of specific interaction of the cells containing HER2 receptors with the aptamers.

Similar kinetics have been observed also for sensing surfaces modified by HeA2_3 aptamers (Appendix A of Appendix A). When sgc8c aptamers were used as non-specific control, the sensor response for the addition of HER2 positive SK-BR-3 cells (5 × 10^4^ cells/mL) has been much lower in comparison with those based on HB5 aptamers (see Figure 5 and Appendix A). We tested also the interaction with the specific sensing surface the HER2 negative cells as a control (MDA-MB-231 cell line). In this case the changes of f_s_ and R_m_ were not significant. The significant changes of f_s_ and R_m_ were observed only when MDA-MB-231 cells were added at the TSM surface modified by sgc8c aptamers which are specific to PTK7 (Appendix A at Appendix A) [20,21]. This is due to the fact that these cells contain PTK7 receptors. In order to exclude non-specific interactions of SK-BR-3 with the NA layer we measured also the values of f_s_ and R_m_ following addition of SK-BR-3 cells (concentration 5 × 10^4^ cells/mL) to a surface of a TSM transducer which contained only an NA layer without aptamers. Addition of the cells resulted in decrease of the f_s_ by 5.0 ± 3.5 Hz and increase of R_m_ by 0.50 ± 0.36 Ω (Appendix A of Appendix A). The change of the frequency was comparable with those of specific interaction (at presence of HB5 aptamers) following addition of 500 cells/mL. Thus, the changes of f_s_ for this aptasensor at non-specific interaction represent 12.1% from whole specific response.

We also compared response of aptasensors formed by different specific aptamers (see Table 1) following addition of HER2 positive SK-BR-3 cells (5x10^4^ cells/mL). The most sensitive response has been obtained for aptasensors prepared by 0.25 µM HB5 and 0.5 µM HeA2_3 aptamers. Figure 5 shows a comparison of the changes of resonant frequency for these aptasensors. The response for an aptasensor formed by not specific aptamers 0.5 µM sgc8c is also shown. It can be seen that higher frequency changes were observed for HeA2_3 aptamers, while much lower changes of frequency were observed for aptasensor prepared by sgc8c aptamers. As it can be seen from Table 1 HeA2_3 aptamer is composed of 40 bases, while HB5 is much longer (85 bases). The constant of dissociation of these aptamers is also different (see Table 1). We suspect that the different responses for these aptasensors can be due to higher affinity of HeA2_3 to HER2 proteins in comparison with those of HB5 as it is revealed from K_d_ values presented in Table 1.

The response of HeA2_3 and HB5 based aptasensors as a function of various concentrations of HER2 positive breast cancer cells SK-BR-3 is summarized on Figure 6 where the calibration curves are presented. The changes in the resonant frequency and motional resistance were detected in a concentration range 50–5 × 10^5^ cells/mL. The changes of the f_s_ and R_m_ values for each concentration were measured on a separate TSM transducer. It can be seen that for both aptasensors f_s_ decreases and R_m_ increases with an increased concentration of the cells. Already at relatively low cell concentration (500 cells/mL) the f_s_ decrease by 4.60 ± 0.08 Hz for aptasensor based on HB5 aptamers. The changes of f_s_ for HeA2_3 based aptasensors were 2 times higher. HB5 aptamer recognizes “HER2 epitope peptide” and HeA2_3 binds to the HER2 protein. It is possible that the HER2 proteins were not been fully available for HB5 aptamers, therefore we obtained less sensitivity of detection in this case. The prognostic concentration of HER2 positive circulating tumor cells (CTCs) in the blood is currently unknown. It seems that only 26.9% of patients with HER2 positive tumor have HER2+ CTCs in their blood. At the same time approx. 24.2% of patients with HER2- tumor has HER+ CTCs in their blood. Therefore the diagnostics of HER2+ CTCs is crucial for the establishment of correct targeted therapy [33]. From the calibration plot the limit of detection (LOD) has been established based on the rule 3.3 × SD/b, where SD is the standard deviation and b is the slope of the regression line, as: 1418 cells/mL for aptasensor based on HeA2_3 aptamer and 1574 cells/mL for aptasensor based on HB5 aptamer.

### 3.4. Amplification of Detection of Breast Cancer Cells by Gold Nanoparticles

Amplification of the detection is important step in the development of high sensitive TSM cell sensors. Gold nanoparticles are often used for this purpose due to their unique electrical and optical properties. They can also be easily modified by receptors such as aptamers or antibodies. Therefore we examined the possibility of amplification of the detection of breast cancer cells by the complexes of gold nanoparticles (AuNPs) conjugated with streptavidin (StpA) and modified by biotinylated HB5 aptamers (HB5-StpA-AuNPs). The interaction of HB5-StpA-AuNPs with the SK-BR-3 cells (concentration 5 × 10^3^ cells/mL) immobilized at the surface of TSM transducer resulted in decrease of resonant frequency, f_s_, by 22.4 ± 3.54 Hz and in an increase of motional resistance, R_m_, by 0.90 ± 0.42 Ω (Figure 7). We tested also addition to a sensor surface of the free aptamers (not conjugated with AuNPs). However, we did not observe significant changes in the measured values until cell concentration of 5 × 10^3^ cells/mL (Table 2). This is due to the much lower weight of the aptamers in comparison with AuNPs. The specificity of the amplification has been tested also by the addition of AuNPs modified by non-specific sgc8c aptamers. In this case the changes of f_s_ and R_m_ values were at the level of the instrument noises. Thus, we have demonstrated the possibility of amplification of the detection of cancer cells by AuNPs modified by aptamers specific to HER2 receptors. In contrast with direct detection of the cells the amplification by nanoparticles improved LOD that reached 550 cells/mL (LOD without amplification it was 1574 cells/mL).

This increase of the sensor sensitivity is not so remarkable. One of the possible reason is the limitation in the penetration depth of ultrasound wave. According to Glassford [34] the penetration depth, δ, of the ultrasound wave from the sensor surface into the water can be calculated according to the equation:δ = (2η/ωρ)^1/2^(1)
where η = 1 mPa.s is viscosity and ρ = 10^3^ kg/m^3^ is density of water at 20 °C, ω is circular frequency (ω = 2πf, f = 8 MHz is the resonant frequency of the quartz crystal). In our case δ = 0.2 μm. This value is less then average thickness of the breast cancer cells, which is in the order of 5–10 μm [35]. Therefore the processes at the cell surface can be detected in less sensitivity by the acoustic method when the thickness of the layer surpass the penetration depth. At the same time the amplification of detection of the thrombin using AuNPs allowing to a decrease of LOD by the factor of 10^2^ [36]. However, monolayers formed by thrombin are much thinner (3–4 nm) in comparison with the cells and are much less than the shear wave penetration depth. Addition of HB5-StpA-AuNPs to the layer of NA-HB5 without cells did not caused any specific interaction and thus no significant changes of frequency and motional resistance were observed (results are not shown).

The changes in resonant frequency due to chemisorption of NA, formation of DNA layer as well as due to adsorption of the cells can be connected with the changes of the mass and shearing viscosity of the layers at TSM transducer. For relatively rigid layers like those formed by NA, from the changes of the resonant frequency it is possible to calculate the mass of the molecular layer using Sauerbrey equation [37]:Δf_s_= −2.26 × 10^−6^ f^2^_0_ (Δm/A)(2)
where f_0_ is the fundamental frequency (8 MHz), Δm is the mass change (in grams) and A = 0.2 cm^2^ is the area of the piezocrystal. The formation of stable NA layer resulted in decrease of resonant frequency by 158 Hz. This corresponds to the mass changes by 218 ng. The number of NA molecules at the surface can be calculated as: (Δm/M_w_)Na = (2.18 × 10^−7^/60,000) × 6.02 × 10^23^ = 2.25 × 10^12^ molecules, where M_w_ = 60,000 g/mol is molecular weight of NA and Na = 6.02 × 10^23^ is Avogadro’s number. In analogy with this calculation it is possible to estimate mass changes and number of molecules after the immobilization of HB5 aptamers and the cells. In the case of aptamers the frequency decrease by 57 Hz corresponds to the mass changes of 78.7 ng. Considering that M_w_ of aptamer is 26 678.5 g/mol, the number of HB5 molecules at the surface is 1.8 × 10^12^, which is comparable with the number of NA molecules. We should, however, note that changes of resonant frequency following immobilization of the aptamers are accompanied also by increase of motional resistance and thus in increase of the contribution of viscous forces. Therefore the viscosity contribution into the mass changes can not be considered as negligible. Thus, the number of immobilized aptamers can be less in comparison with those determined based on the Sauerbrey equation.

Taking into account that the NA molecule has 4 binding sites for biotin, but 2 are not available due to chemisorption, a maximum of 2 binding sites can be occupied by aptamers. From the above rough estimation it is, however, clear that the NA surface is not maximally saturated by aptamers. This result is not unexpected, considering that DNA aptamers form 3D structures in a water solution, have certain volume and due to negative charge cannot be too close to each other at the surface. Therefore they cannot occupy all NA binding sites.

Addition of breast cancer cells at the concentration of 5 × 10^4^ cells/mL at the sensing surface resulted in a frequency decrease by 46.7 Hz. This corresponds to the increase of the mass by 64.5 ng. The weight of one cell is approx. 1.5 to 3.5 ng [38]. This means that number of adsorbed cells is around 18 to 43. However, considering also the viscosity contribution into the frequency the estimated number of cells is even less. At the same time the estimated decay length for shear wave in water solution is only 0.2 μm. Therefore only a small fraction of the cell mass is detected by the frequency shift. This result agree well with those reported by Fredriksson et al. [39].

In the current literature there are only a few studies focused on the detection of cancer cells using the acoustics TSM method. The proposed cell sensors are characterized by different values of detection limits. In particularly, Pan et al. [40] developed an aptasensor for detection of an acute leukemia cell line CCRF-CEM using magnetic nanoparticles conjugated with DNA aptamers. The LOD of this aptasensor was 8 × 10^3^ cells/mL and the linear range was 10^4^–1.5 × 10^5^ cells/mL. Shan et al. [41] developed TSM aptasensor for detection the CCRF-CEM cell lines with LOD of 1160 cells/mL. Better LOD for detection leukemics Jurkat and MOLT-4 cells (195–570 cells/mL) has been reported in our recent paper [19].

Only one paper has been so far reported on the development of TSM biosensor for detection breast cancer cells. Zhang et al. [42] reported high sensitive cell sensor for detection of breast cancer cell lines MCF-7. The TSM transducer has been modified by chitosan and folic acid. The folic acid can specifically recognize the folic acid receptors that were over expressed at MCF-7 cells. The LOD for this sensor was 430 cells/mL, which is comparable with our TSM aptasensor after amplification of its response by AuNPs. Detection of SK-BR-3 HER2 positive cancer cells has also been reported [35]. But for this purpose an sandwich assay composed of monoclonal anti-HER2 antibodies immobilized at glassy carbon electrode surface with adsorbed AuNPs has been used for interaction with the SK-BR-3 cells. Then the AuNPs modified by RNA aptamers specific to HER2 receptors were used. The silver ions reduced by hydrazine were deposited to the AuNPs and detected by square wave stripping voltammetry. Using this electrochemical sensor it was possible to reach rather sensitive LOD of 26 cells/mL. The aptasensor for detection free HER2 proteins has also been reported [2]. For this purpose the electrochemical impedance spectroscopy (EIS) has been used. The HER2 specific DNA aptamers were immobilized at the gold nanoparticles deposited at the surface of gold electrode. EIS method allowed detection of HER2 proteins in a concentration range of 10^−^^5^ to 10^2^ ng/mL.

Despite the fact that our acoustics aptasensor is less sensitive, it represent good alternative to the electrochemical sensors. First, the acoustics response can be easily and directly monitored without any labels. This technique can be easily operated. Relatively low cost TSM devices are currently available at the market. In addition, by optimalization of the aptamer immobilization using nanostructures such as carbon nanotubes, dendrimers, etc. it is possible to increase of aptamer surface density which may improve the detection limit.

## 4. Conclusions

In this work we demonstrated label-free detection of HER2 positive SK-BR-3 breast cancer cells using thickness shear mode acoustics method (TSM). The sensor based on immobilization of biotinylated DNA aptamers adsorbed at the neutravidin self-assembled monolayer at the gold surface of TSM transducer allowed specific detection of the cancer cells with limit of detection (LOD) of 1574 cells/mL (HB5 aptamer based sensor) or 1418 cells/mL (HeA2_3 aptamer based sensor). The amplification of detection using gold nanoparticles modified by aptamers allowed to improve detection limit only up to 550 cells/mL. This is due to the fact that the thickness of the cell layer is much higher (5–10 μm) in comparison with shear wave penetration depth (0.2 μm). Therefore the decrease in the frequency observed reflects only small part of the whole cell mass. Thus, further improvement of the sensitivity of detection by acoustics method can be obtained by optimization of aptamer immobilization using nanostructures such as carbon nanotubes, dendrimers or other nanomaterials. We believe that using this approach further progress in acoustics cell sensors will be possible toward practical application in the cancer diagnostics in particularly based on the detection of circulating tumor cells (CTC).

## Figures and Tables

**Figure 1 biosensors-09-00072-f001:**
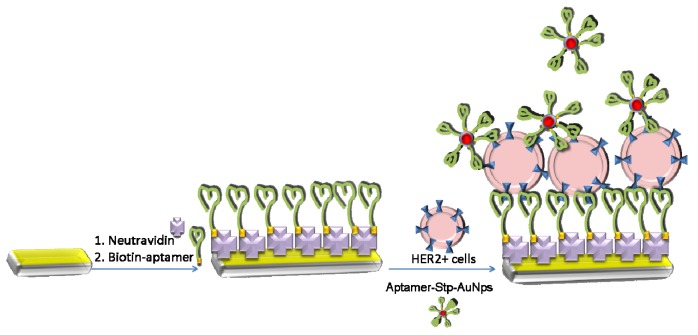
The scheme of the preparation of the aptasensor and binding of the cells at the aptamer surface. The gold nanoparticles modified by specific aptamers are used for the amplification of the cell detection.

**Figure 2 biosensors-09-00072-f002:**
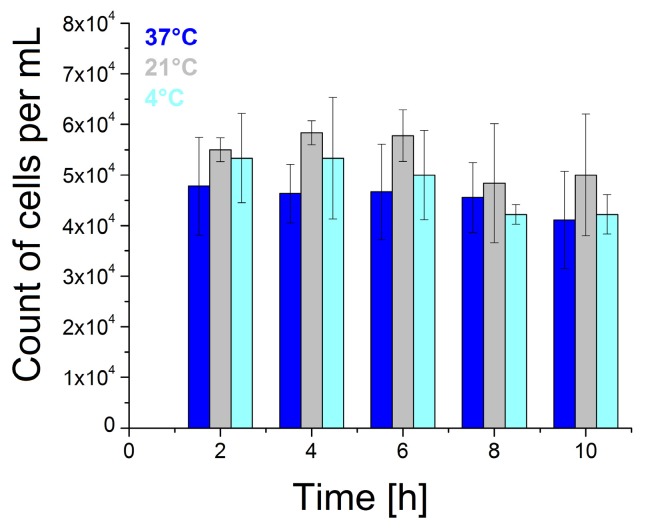
Viability of the SK-BR-3 cells in PBS in the incubator (37 °C), at laboratory temperature (20 °C) and in refrigerator (4 °C). The data represent mean ± SD obtained from 3 independent experiments in each series.

**Figure 3 biosensors-09-00072-f003:**
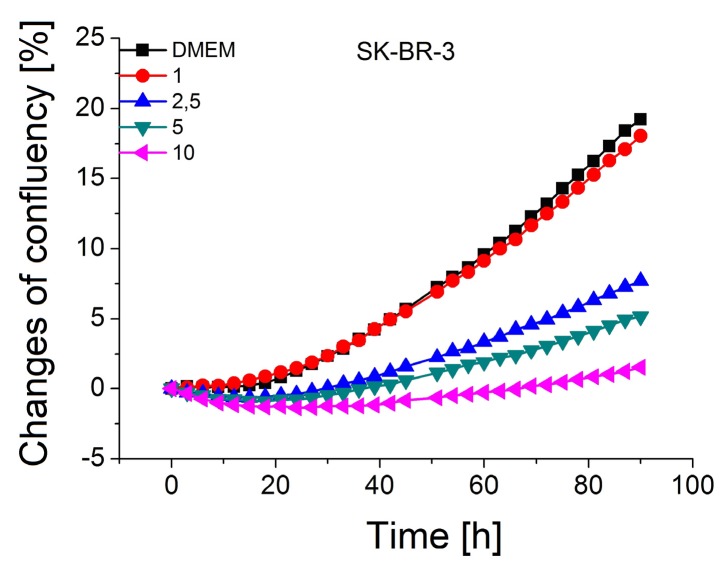
Proliferation assay of SK-BR-3 cells in DMEM medium and after exposure to streptavidin-modified AuNPs (1–10 µg/mL) in PBS. The concentration of nanoparticles in μg/mL is shown at inset.

**Figure 4 biosensors-09-00072-f004:**
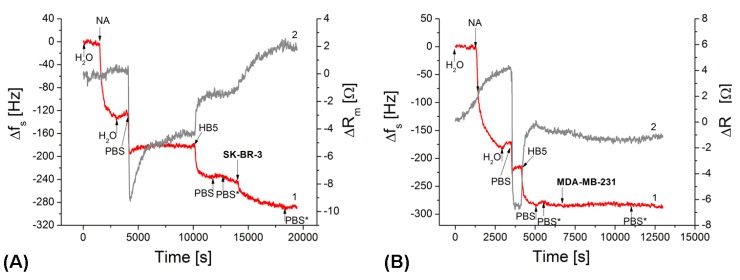
The kinetics changes of the series resonant frequency, Δf_s_, (curve 1) and motional resistance, ΔR_m_, (curve 2) of TSM transducer following addtion of NA (125 μg/mL), HB5 aptamers (0,25 μM) and SK-BR-3 cells (**A**) or MDA-MB-231 cells (**B**). The concentration of the cells was 5 × 10^4^ cells/mL. The arrows indicate addition of respective compounds as well as washing of the surface by water, PBS and by PBS with addition of aliquote volume of DMEM (PBS*).

**Figure 5 biosensors-09-00072-f005:**
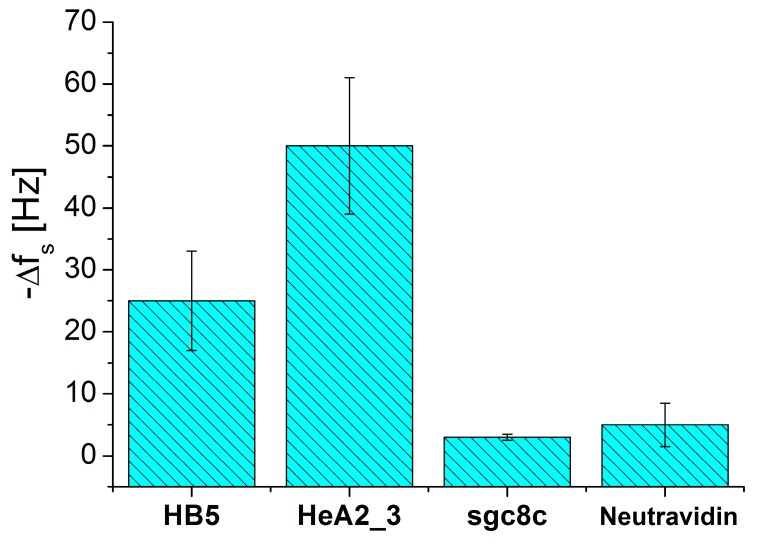
Changes of the resonant frequency for aptasensors formed by HB5, HeA2_3, sgc8c aptamers and without aptamer (only neutravidin layer was presented) following interaction with HER2 positive SK-BR-3 cells (5 × 10^4^ cells/mL). Results are mean ± SD obtained from 3 independent experiments in each series.

**Figure 6 biosensors-09-00072-f006:**
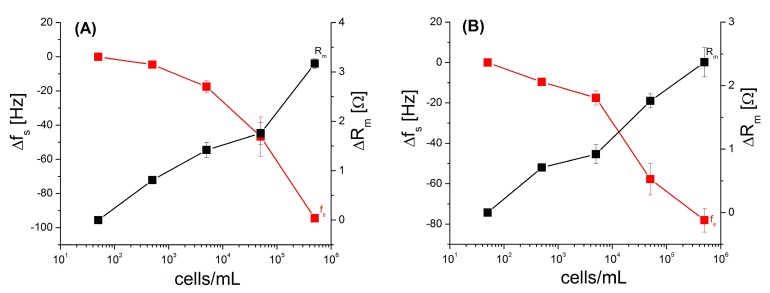
The plot of the Δf_s_ and ΔR_m_ as a function of the concentration of HER2+ cells SK-BR-3 for aptasensors based on HeA2_3 (**A**) and HB5 (**B**) aptamers. Results represent mean ± SD obtained from 3 independent experiments in each series.

**Figure 7 biosensors-09-00072-f007:**
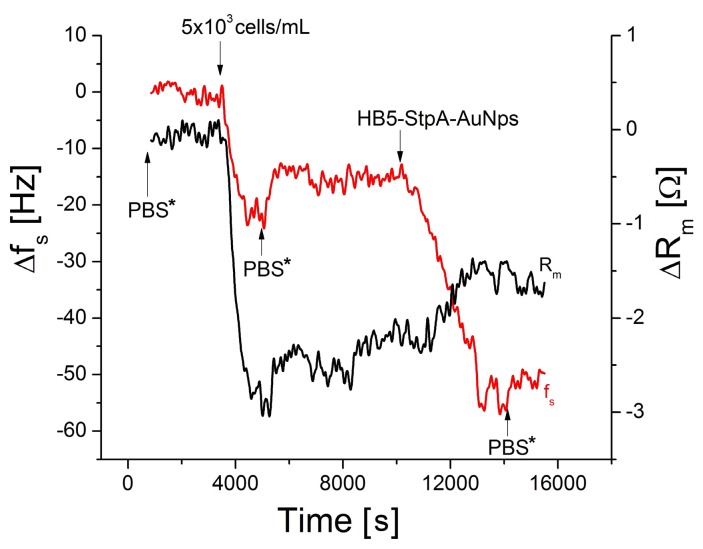
The changes of resonant frequency, Δf_s_, (red line) and motional resistance, ΔR_m_, (black line) of TSM transducer modified by HB5 aptamers following addition of SK-BR-3 cells (5 × 10^3^ cells/mL) and the HB5-StpA-AuNPs complexes. The arrows indicate the addition of respective compounds and washing of the surface by PBS+DMEM solution (PBS*).

**Table 1 biosensors-09-00072-t001:** Sequences of DNA aptamers used in experiments. K_d_ is the dissociation constant.

Target Protein	Aptamer	Sequence 5′-3′	K_d_, nM	Reference
Epitope peptide HER2	HB5	AACCGCCCAAATCCCTAAGAGTCTGCACT TGTCATTTTGTATATGTATTTGGTTTTTGGC TCTCACAGACACACTACACACGCACA	18.9	[13]
HER2 protein	HeA2_3	TCTAAAAGGATTCTTCCCAAGGGGATCCA ATTCAAACAGC	6.2	[11]
PTK7 protein	Sgc8c	ATCTAACTGCTGCGCCGCCGGGAAAATAC TGTACGGTTAGA	1.0	[20,21]

**Table 2 biosensors-09-00072-t002:** Comparison of the changes of resonant frequency (Δf_s_) and motional resistance (ΔR_m_) of TSM transducer after addition of 0.25 µM HB5 aptamers and/or gold nanoparticles modified by HB5 aptamers (HB5-StpA-AuNPs) (concentration 1 μg/mL) to a surface of HB5 based aptasensor with the immobilized breast cancer cells (SK-BR-3) of various concentrations. Results are mean ± SD obtained from 3 independent experiments in each series.

Cells/mL	HB5 Aptamer		Complex HB5-StpA-AuNPs	
	Δ*f_s_* ± SD (Hz)	Δ*R_m_* ± SD (Ω)	Δ*f_s_* ± SD (Hz)	Δ*R_m_* ± SD (Ω)
50	0	0	0	0
5 × 10^2^	0	0	−3.5 ± 3.25	0.4 ± 0.10
5 × 10^3^	−1 ± 4.2	0 ± 0.30	−22.4 ± 3.54	0.9 ± 0.42
5 × 10^4^	−7.3 ± 3.54	1.17 ± 0.38	−45.9 ± 4.50	1.8 ± 0.54

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
