# Peer review of "Detection of Breast Cancer Cells Using Acoustics Aptasensor Specific to HER2 Receptors"

_biosensors, 2019, doi:10.3390/bios9020072_

Reviewer 1 Report

The article describes interesting results with good experiment design. Here are some questions to be adressed prior acceptance.

1) Please clarify why citotoxity study is really required? If even cells do not grow with higher concentration of nanoparticles, how can it impair the results of acoustic analysis if the cells seem not to grow anymore on the sensor surface?

2) Is there any reason dehind using HB5 aptamer for LoD enhancement while HeA2_3 showed better sensing results? 

3) Fig4 shows interaction of HB5 aptamer with SK-BR-3 cells. The same graphs for control HER2-negative cells and for other aptamers with both cell lines are requires and can be either added to Fig4 or shown as Supplementary material. 

4) Fig5 should be supplemented with a bar for non-scecific response of NA-covered sensor described in lanes 326-333.

5) Please, explain what was refered to as noise signal in LoD  caclulations (see lanes 367-369),  and specify what value was used for its calculation. Was it Λfs?

6) Are there any data known about LoD for practically implemented methods for HER2 detection (e.g. FISH mentioned in the introduction)? Or may be these methods can be compared to the developed biosensor with some other parameters in the Discussion section?

Author Response

We are grateful to this reviewer for useful comments that allowed us to improve manuscript.

Comment 1: Please clarify why cytotoxity study is really required? If even cells do not grow with higher concentration of nanoparticles, how can it impair the results of acoustic analysis if the cells seem not to grow anymore on the sensor surface?

Response: The reviewer is right that the cells in a flow system of TSM do not grow. However, it is crucial to provide viability of the cells during whole experiment as well as following addition of nanoparticles. As it follows from the literature the gold nanoparticles can be toxic for the cells. It was therefore important to analyze what concentration of nanoparticles is not toxic and do not induce cell apoptosis and that  the number of the cells during experiment is stable. We included this explanation in the revised version, page 6, part 3.2 (lines 241-244).

Comment 2: Is there any reason behind using HB5 aptamer for LoD enhancement while HeA2_3 showed better sensing results?

Response: The main reason of the applications of two different aptamers for detection breast cancer cells was as follows. Both aptamers differ in their length as well as in the secondary and 3D structures, which can affect their interactions with HER2 receptors at the cell surface. HB5 aptamer recognizes “HER2 epitope peptide” and  HeA2_3 binds to the HER2 protein. It is possible that the HER2 proteins were not been fully available for HB5 aptamers, therefore we obtained less sensitivity of detection in this case. We included this explanation in revised text, page 9 (lines 370-372).

Comment 3: Fig. 4 shows interaction of HB5 aptamer with SK-BR-3 cells. The same graphs for control HER2-negative cells and for other aptamers with both cell lines are requires and can be either added to Fig. 4 or shown as Supplementary material.

Response: We agree with the reviewer's comment and added requested figures into the section Supplementary material (Part S1).

Comment 4: Fig. 5 should be supplemented with a bar for non-specific response of NA-covered sensor described in lines 326-333.

Response: We agree with this comment and improved figure 5 accordingly.

Comment 5: Please, explain what was referred to as noise signal in LoD  calculations (see lines 367-369),  and specify what value was used for its calculation. Was it Δfs?

Response: The LOD has been calculated based on the changes of the frequency according to equation LOD = (3.3xSD)/b, where SD is standard deviation, b is slope of the regression line. For HB5 and HeA2_3 aptamers we obtained following LOD values:

HB5: (3.3x2.48)/0.0052 = 1574 cells/mL

HeA2_3: (3.3x1.16)/0.0027 = 1418 cells/mL

Comment 6: Are there any data known about LoD for practically implemented methods for HER2 detection (e.g. FISH mentioned in the introduction)? Or may be these methods can be compared to the developed biosensor with some other parameters in the Discussion section?

Response: In the FISH method the analysis of  HER2 overexpression is based on the determination of corresponding nucleotide sequences (Koudelakova et al. J. Mol. Diagn. 17 (2015) 446-455). The DNA is isolated from the cell tissue, amplified by PCR and detected by FISH method. Unfortunately it is principally different from determination of the HER2 positive cells. Therefore LOD for these two methods can not be compared.

Reviewer 2 Report

The paper describes the use of an aptasensor using thickness shear mode acoustics method for the detection of SK-BR-3 breast cancer cells.  Although the paper would be of interest to the readership of the journal, I have real doubts about the preparation of the sensor surface which I have detailed below.  I therefore cannot recommend publication.

The authors should take note of the following points.

The authors should get their manuscript proofread by a native speaker as I am detecting a large number of grammatical errors. 

For example “The Aptamer” rather than “aptamers”.

Line 62:  A low pH would result in depurination of the DNA aptamer?

Line 64:  This is not correct.  SPR sensors which use antibodies can be regenerated without loss of affinity of the antibody.

Figure 1: At the surface of What?  This figure should be expanded to show the role of gold nanoparticles in the assay design.

Was a control signal measured?  This would give a better indication as to whether you have binding with the aptamer. 

Was the aptamer refolded before use?

Line 190:  The assumption that cysteine residues are adsorbed on the gold is incorrect here.  Most cysteine residues exist as a disulphide bridges and it is doubtful that there are any free SH groups free to form interactions with the gold.  I´m not sure whether the authors are implying that a covalent bond is formed between the gold and SH groups.  I would expect that avidin would form strong non-covalent interactions with the gold due to its isoelectronic point.  However neutravidin has a neutral Pi value and as such would have minimal non-specific interactions with the gold.  This makes me very sceptical that the authors have indeed successfully attached their aptamers to the gold.  I´m even more doubtful that the neutravidin forms a SAM as that would require an ordered self-assembly of the protein.  I would suspect that there are gaps between each neutravidin protein on the gold and hence large non-specific interactions.  Therefore the authors need to demonstrate the characterisation of neutravidin to the surface more convincingly. I also suspect that the neutravidin can be easily washed away limiting the robustness of the sensor.  The authors should demonstrate the reusablility and robustness of the sensor.  I would have suggested that using a thiol SAM and attaching the neutravidin via an amine coupling would be much better protocol.

Author Response

We are grateful to this reviewer for useful comments that allowed us to improve manuscript.

Comment: The authors should get their manuscript proofread by a native speaker as I am detecting a large number of grammatical errors. For example “The Aptamer” rather than “aptamers”.

Response: Thank you for the comment. The manuscript has been improved accordingly.

Comment: Line 62:  A low pH would result in depurination of the DNA aptamer?

Response: The depurination of nucleic acids can certainly takes place at low pH. However, it has been shown that this process is accelerated only at very low pH (below pH 3) (see for example An et al. Plos One 9 (2014) e115950). However, our study of the interaction of aptamers with target molecules at various pH showed that already at pH 4.5 the aptamers lost the ability of binding the proteins (Hianik et al. Bioelectrochemistry 70 (2007) 127–133). Therefore, at this pH value the regeneration is possible without significant DNA depurination. We modified the text accordingly. (Lines 62-64).

Comment: Line 64:  This is not correct.  SPR sensors which use antibodies can be regenerated without loss of affinity of the antibody.

Response: We agree with the reviewer and modified corresponding text as follows:

"Moreover, in contrast with antibodies the aptamers are more stable in respect of temperature denaturation-renaturation and the sensor based on aptamers can be regenerated in certain conditions, such as for example high ionic strength [10]. (lines 62-64).

Comment: Figure 1: At the surface of What?  This figure should be expanded to show the role of gold nanoparticles in the assay design.

Response: We are grateful for this comment and modified Figure 1 accordingly.

Comment: Was a control signal measured?  This would give a better indication as to whether you have binding with the aptamer.

Response: Yes, we studied in detail the interaction of the cells with non-specific aptamers as well as with the sensing surface covered only by neutravidin, without aptamers. The corresponding figures are presented now in Supplementary material section. In addition, it has been shown that adherent breast cancer cells can be non-specifically adsorbed to a gold surface, however, washing by PBS fully removed these cells from the surface, which correspond the conditions used in our work (Wegener et al. Biochem. Biophys. Meth. 32 (1996) 151-170).

Comment: Was the aptamer refolded before use?

Response: Yes, we analyzed sensor response following refolding of aptamers. For this purpose the aptamer stock solution has been heated up to 95 oC and then slowly cooled until room temperature (20 oC). However, surprisingly, aptamer refolding resulted in lower sensitivity of the sensor in comparison with immobilization of aptamers without this step. Therefore we prepared aptasensors without preliminary refolding.

Comment: Line 190: The assumption that cysteine residues are adsorbed on the gold is incorrect here.  Most cysteine residues exist as a disulphide bridges and it is doubtful that there are any free SH groups free to form interactions with the gold.  I am not sure whether the authors are implying that a covalent bond is formed between the gold and SH groups.  I would expect that avidin would form strong non-covalent interactions with the gold due to its isoelectronic point.  However neutravidin has a neutral Pi value and as such would have minimal non-specific interactions with the gold.  This makes me very skeptical that the authors have indeed successfully attached their aptamers to the gold.  I´m even more doubtful that the neutravidin forms a SAM as that would require an ordered self-assembly of the protein.  I would suspect that there are gaps between each neutravidin protein on the gold and hence large non-specific interactions.  Therefore the authors need to demonstrate the characterization of neutravidin to the surface more convincingly. I also suspect that the neutravidin can be easily washed away limiting the robustness of the sensor.  The authors should demonstrate the reusability and robustness of the sensor.  I would have suggested that using a thiol SAM and attaching the neutravidin via an amine coupling would be much better protocol. 

Response: Thank you for this comment. However, the application of neutravidin for the formation of stable layer at the gold has been demonstrate earlier in the works by Thompson et al (for example Tassew and Thompson, Biophys. Chem. 106 (2003) 241-252). Also we confirmed this by our SPR study, that demonstrated the stability of the neutravidin layers at gold surface with subsequently immobilized biotinylated aptamers (Ostatna et al. Anal. Bioanal. Chem. 391 (2008) 1861-1869). In order to prove this concept we also performed detailed AFM study of the topography of the neutravidin at the gold surface. The scratching experiment clearly demonstrated stability of neutravidin layer. These results have been include in Supplementary material of the revised manuscript  (Part S2).

Reviewer 3 Report

1. I am interested in specificity data as well as repeatability/reproducibility of the sensor performance.

2. Please provide analytical soundness of the sensor ?

3. Did author notice any non specific adsorption of any substance on electrode surface?

4. What about analysis time of the method?

5. Please include sensitivity in the abstract?/

Author Response

We are grateful to this reviewer for useful comments that allowed us to improve manuscript.

Comment 1: I am interested in specificity data as well as repeatability/reproducibility of the sensor performance.

Response: We confirmed the specificity of sensor response also by using non-specific aptamers as well as control cells that does not contain HER2 positive receptors. The reproducibility has been confirmed by repeating the experiments at the same conditions for 3-5 times. Additional data are presented in Supplementary material section.

Comment 2:Please provide analytical soundness of the sensor ?

Response: The results obtained can be useful for further development of the aptasensors for detection of circulating tumor cells (CTC).

Comment 3: Did author notice any non specific adsorption of any substance on electrode surface?

Response: Yes, the component of the cultivation media, mostly FBS resulted non-specific interaction with the sensor surface. Therefore the sensing surface has been washed by PBS containing aliquot concentration of cultivation media DMEM.

Comment 4. What about analysis time of the method?

Response: Whole measurement that included preparation of the sensing layer and addition of the cells had duration of around 6 hours as it is evidence from kinetics results (see for example Fig. 4 as well as  figures at Supplementary material section).

Comment 5: Please include sensitivity in the abstract?

Response: The sensitivity of breast cancer cell detection (limit of detection) has been included in the abstract of revised manuscript.

Round  2

Reviewer 2 Report

The authors have addressed all the points raised in the first review.  I, therefore, recommend publication after further proofreading of the manuscript.